# Calibration and Cross-Validation of Accelerometer Cut-Points to Classify Sedentary Time and Physical Activity from Hip and Non-Dominant and Dominant Wrists in Older Adults

**DOI:** 10.3390/s21103326

**Published:** 2021-05-11

**Authors:** Jairo H. Migueles, Cristina Cadenas-Sanchez, Juan M. A. Alcantara, Javier Leal-Martín, Asier Mañas, Ignacio Ara, Nancy W. Glynn, Eric J. Shiroma

**Affiliations:** 1Department of Health, Medicine and Caring Sciences, Linköping University, 581 83 Linköping, Sweden; jairo.hidalgo.migueles@liu.se; 2PROFITH “PROmoting FITness and Health Through Physical Activity” Research Group, Department of Physical Education and Sports, Faculty of Sport Sciences, Sport and Health University Research Institute (iMUDS), University of Granada, 18010 Granada, Spain; cristina.cadenas.sanchez@gmail.com (C.C.-S.); alcantarajma@ugr.es (J.M.A.A.); 3Institute for Innovation & Sustainable Development in Food Chain (IS-FOOD), Public University of Navarra, 31006 Pamplona, Spain; 4GENUD Toledo Research Group, Universidad de Castilla-La Mancha, Avenida Carlos III s/n, 45071 Toledo, Spain; javier.leal@uclm.es (J.L.-M.); asier.manas@uclm.es (A.M.); ignacio.ara@uclm.es (I.A.); 5CIBER of Frailty and Healthy Aging (CIBERFES), 28029 Madrid, Spain; 6Department of Epidemiology, Graduate School of Public Health, University of Pittsburgh, Pittsburgh, PA 15260, USA; epidnwg@pitt.edu; 7Laboratory of Epidemiology and Population Science, Intramural Research Program of the National Institutes of Health, National Institute on Aging, Baltimore, MD 20892, USA

**Keywords:** sedentary behavior, movement, light intensity, exercise, sitting, elderly, counts, ENMO

## Abstract

Accelerometers’ accuracy for sedentary time (ST) and moderate-to-vigorous physical activity (MVPA) classification depends on accelerometer placement, data processing, activities, and sample characteristics. As intensities differ by age, this study sought to determine intensity cut-points at various wear locations people more than 70 years old. Data from 59 older adults were used for calibration and from 21 independent participants for cross-validation purposes. Participants wore accelerometers on their hip and wrists while performing activities and having their energy expenditure measured with portable calorimetry. ST and MVPA were defined as ≤1.5 metabolic equivalents (METs) and ≥3 METs (1 MET = 2.8 mL/kg/min), respectively. Receiver operator characteristic (ROC) analyses showed fair-to-good accuracy (area under the curve [AUC] = 0.62–0.89). ST cut-points were 7 m*g* (cross-validation: sensitivity = 0.88, specificity = 0.80) and 1 count/5 s (cross-validation: sensitivity = 0.91, specificity = 0.96) for the hip; 18 m*g* (cross-validation: sensitivity = 0.86, specificity = 0.86) and 102 counts/5 s (cross-validation: sensitivity = 0.91, specificity = 0.92) for the non-dominant wrist; and 22 m*g* and 175 counts/5 s (not cross-validated) for the dominant wrist. MVPA cut-points were 14 m*g* (cross-validation: sensitivity = 0.70, specificity = 0.99) and 54 count/5 s (cross-validation: sensitivity = 1.00, specificity = 0.96) for the hip; 60 m*g* (cross-validation: sensitivity = 0.83, specificity = 0.99) and 182 counts/5 s (cross-validation: sensitivity = 1.00, specificity = 0.89) for the non-dominant wrist; and 64 m*g* and 268 counts/5 s (not cross-validated) for the dominant wrist. These cut-points can classify ST and MVPA in older adults from hip- and wrist-worn accelerometers.

## 1. Introduction

The benefits of physical activity (PA) in the elderly are well established [1,2]. Accelerometers, wearable devices that continuously capture movement, have been used to objectively measure PA. Older adults often show high recall bias in self-reports, which can be overcome by accelerometers [3]. Modern accelerometers provide high-frequency accelerations (often in *G*s*,* 1 *G* ~ 9.8 m/s^2^) in three axes. Calibration studies are crucial to derive cut-points that translate accelerations into sedentary time (ST) or PA intensities (e.g., light, moderate-to-vigorous (MVPA)) based on metabolic equivalents of the tasks (METs, the ratio of the energy expended during an activity to the resting metabolic rate) [4,5,6].

These cut-points are population specific, as both PA patterns and METs are highly modifiable by physiological factors (e.g., age, sex, cardiorespiratory fitness). Furthermore, cut-points are protocol specific, being dependent on the accelerometer body attachment site or the activities performed in the calibration study, among others. As such, although a number of cut-points have been derived for older adults [5,6,7], different populations and protocols make difficult the application of these cut-points in other samples. The rapid advancement of the PA measurement field in the past decade, mainly driven by the access to accelerometer raw data, has resulted in a myriad of methods to process these data [8]. Studies deriving consistent cut-points from different body attachment sites and using different processing protocols are lacking. The development of consistent cut-points across device locations and protocols and the cross-validation of cut-points are crucial steps to ensure data replicability and representativeness across the field [9].

Therefore, this study aimed to (i) derive accelerometer cut-points to classify ST and MVPA from accelerometers attached to the hip, non-dominant wrist, and dominant wrist in people more than 70 years old and (ii) cross-validate the derived cut-points in an independent sample of older adults performing different activities.

## 2. Materials and Methods

### 2.1. Study Design

The Aging Research Evaluating Accelerometry (AREA) study, part of the Developmental Epidemiologic Cohort Study [10] conducted at the University of Pittsburgh (USA), is a methodological study designed to examine the impact of the accelerometer wear location on ST and PA assessment in older adults. In the laboratory protocol, participants wore three GT3X+ accelerometers (ActiGraph, Pensacola, FL, USA) placed on the right hip and left and right wrists. The study was approved by the institutional review board of the University of Pittsburgh and the National Institute on Aging; all participants provided written informed consent.

Additionally, data from the MOvement and BEhaviours MeasuremENT (MOBEMENT) study conducted at the University of Castilla-La Mancha (Spain) were used for cross-validation purposes. The MOBEMENT study is an ongoing project that intends to derive cut-points for older adults with differing physical function and frailty status. As of 2020, it collected data of 21 participants, which will be used to cross-validate the findings of this study. The MOBEMENT study was approved by the ethical committee of the Toledo Hospital Complex, and written informed consent was obtained from participants.

### 2.2. Participants

A total of 87 older adults (≥70 years old) participated in the AREA study. Of these, 59 participants were analyzed after excluding participants for accelerometer calibration issues (*n* = 13), accelerometer recording data failings (*n* = 3), indirect calorimeter (metabolic cart) calibration issues (*n* = 4), or problems with data synchronization (*n* = 8). Regarding the MOBEMENT study, data from 21 older adults (≥65 years old) were used. Two accelerometers were not correctly calibrated, resulting in exclusion; thus, 19 participants were included for cross-validation purposes. All these 19 participants were ≥70 years old.

### 2.3. Procedures

Both in the AREA and in the MOBEMENT study, participants completed a laboratory-based protocol simulating activities of daily life. Activity descriptions and durations are provided in Table 1. Older adults mainly engage in ST and light-intensity activities, such as slow walking, carrying of light objects, and household chores. As such, activities in the laboratory protocol intended to be representative of these PA patterns.

In the AREA study, activity durations were selected not to induce fatigue but to reach a steady state in the measured gas exchange, concretely in the volume of oxygen consumption (VO_2_) and volume of carbon dioxide production (VCO_2_). However, some of the activities were based on the time the participants required to complete a certain task. In these cases, some activities were not long enough to accurately determine METs and subsequently were excluded from the analyses. This is the case of chair stands and the 20 m walk activities. Regarding the MOBEMENT study, all activities had a duration of 5 min, except for the 6 min walking test. In this test, participants were asked to walk as fast as possible in order to cover the maximum distance that they could in 6 min.

### 2.4. Measures

#### 2.4.1. Anthropometrics

The participants’ body mass and standing height were measured with an electronic scale and standing stadiometer, respectively. The body mass index was calculated as body mass (kg) divided by squared height (m^2^).

#### 2.4.2. Accelerometers

Participants from the calibration sample (AREA study) wore three ActiGraph GT3X+ accelerometers on the right hip and the dominant and non-dominant wrists. Participants from the cross-validation sample (MOBEMENT study) wore two ActiGraph GT3X+, one on the right hip and one on the non-dominant wrist. The GT3X+ is a watch-sized lightweight device able to collect accelerations at 30–100 Hz within a dynamic range of ±8 *G*s in the three axes: vertical, antero-posterior, and medio-lateral. Devices were set to collect accelerations at 80 Hz in the AREA study and 60 Hz in the MOBEMENT study. After testing, the accelerometer raw data were downloaded in .gt3x and .csv formats for data processing. Second-by-second acceleration data were generated in the GGIR R package [11] and ActiLife software (ActiGraph, Pensacola, FL, USA).

The GGIR R package version 2.0.0 [11] facilitated the data cleaning and extraction of acceleration levels as Euclidean Norm Minus One *G* (ENMO), with negative values rounded to zero. ENMO has been previously used to derive accelerometer cut-points for older adults [6] and has demonstrated to be comparable across different accelerometers [6,12]. ENMO values were calculated on a second-by-second basis and subsequently averaged over 5 s epochs for analyses. Actilife software was used to calculate activity counts, an acceleration metric intended to measure body movement by filtering out accelerations outside 0.25–2.5 Hz. Activity counts were calculated for every axis, and then the vector magnitude was calculated every second and finally aggregated in 5 s epochs.

The same procedures were followed for the hip- and wrist-worn accelerometers, and values from the different accelerometers were synchronized. To calculate the acceleration values for each activity, the first and last seconds of each activity were excluded to ensure data stability. Overall, around 3 min were used to calculate the acceleration values of each activity (see specific times in Table 1).

#### 2.4.3. Energy Expenditure

The VO_2_ and VCO_2_ (in mL/min) gas exchange was measured with a portable indirect calorimetry system (COSMED k4b^2^ in the AREA study and COSMED k5 in the MOBEMENT study; COSMED s.r.l., Rome, Italy). COSMED k4b^2^ is a lightweight (925 g), self-contained gas analyzer valid for the measurement of gas exchange [13]. Gas exchange was measured breath by breath and then matched to accelerometer second-by-second timestamps. Absolute METs were calculated by dividing the VO_2_ at a steady state by the body mass and by 2.8, which is the average value of 1 MET in older adults [14]. The METs associated with each activity were obtained from averaging the last 2 min after removing the peaks in the VO_2_ measurement (i.e., peaks deviating >2 SD from the mean, i.e., potential artifacts) (Table 1). The decision of dividing by 2.8 (instead of the usual value for adults, 3.5 mL/kg/min) was supported by calculating the individual value of 1 MET using the data from the first activity performed in the AREA study (i.e., lying still for 10 min) and a specific protocol in the MOBEMENT study (i.e., lying still for 30 min). On average, the observed resting metabolic rate (1 MET) was 3 (SD = 0.7) mL/kg/min in the AREA study participants and 2.7 (SD = 0.7) mL/kg/min in the MOBEMENT study participants. This measure was obtained from averaging the last 5 min of the lying-still activity after removing peaks in the VO_2_ signal. Since longer gas exchange assessments protocols are recommended to assess the resting metabolic rate [15], this value might be overestimated, which agrees with the expected decrease in the resting metabolic rate associated with aging [14].

### 2.5. Statistics

Descriptive characteristics of AREA and MOBEMENT study participants were calculated as means and standard deviations. Unpaired *t*-tests were used to compare demographic factors between AREA and MOBEMENT study participants and between men and women in each study. All statistical analyses were performed in R v 4.0.3.

#### 2.5.1. Cut-Point Calibration

Participants (*n* = 59; 30 women) from the AREA study provided calibration data. Descriptive statistics for both acceleration metrics (ENMO and counts) were calculated for each activity performed in the calibration protocol. MET values were used to classify each activity as ST (≤1.5 METs) or MVPA (≥3 METs) as it is standard in older adults when the value of 1 MET has been adapted to this population [6,16,17]. Dichotomous variables were then obtained for ST and MVPA occurrences, with 1 representing the behavior of interest versus 0 representing other behaviors. Receiver operating characteristic (ROC) curves were used to determine ST-to-light PA and light-to-MVPA cut-points [16]. The pROC R package was used for ROC analyses [17]. The area under the curve (AUC) was obtained from the sensitivity versus specificity curves as a measure of diagnostic accuracy for each threshold derived, and 95% confidence intervals (CI_95%_) for the AUCs were derived [18]. AUC values of ≥0.90 were considered excellent; 0.80–0.89, good; 0.70–0.79, fair; and <0.70, poor [19]. The accuracy to identify ST and MVPA using different acceleration metrics (ENMO vs. counts) or different body attachment sites (hip vs. non-dominant wrist vs. dominant wrist) were compared with non-parametric tests to investigate the differences in the AUCs by the different ROC curves performed [18]. Then, thresholds were developed, seeking to maximize both sensitivity (i.e., ability to identify that the behavior of interest is occurring (true positives)) and specificity (ability to identify that the behavior of interest is not occurring (true negatives)). The aim was to correctly classify ST and MVPA (sensitivity), while limiting the misclassification of ST and MVPA (specificity). As such, we looked for the closest threshold to the perfect sensitivity and specificity, i.e., the minimum value in Equation (1) [20]:(1 − sensitivities)^2^ + (1 − specificities)^2^(1)

#### 2.5.2. Cut-Point Cross-Validation

Thresholds derived in the calibration sample were cross-validated in an independent sample performing a different protocol of activities [9]. We cross-validated the set of cut-points developed in the calibration sample by comparing them to MET values in the cross-validation sample (instead of developing a new set of cut-points). The cross-validation sample comprised 19 participants from the MOBEMENT study (10 women). ST and MVPA were identified and categorized into dichotomous variables replicating the calibration analyses. ROC curves were performed, and sensitivity and specificity values for the thresholds derived in the calibration analyses were obtained.

## 3. Results

Descriptive characteristics of the calibration (AREA) and cross-validation (MOBEMENT) participants are presented in Table 2. No significant differences in age or the BMI were found between men and women in any of the samples (*p* > 0.35). Age, sex, and anthropometrics were not statistically different between the participants included (*n* = 59) and excluded (*n* = 30) from the AREA study (Table A1, Appendix B). The cross-validation sample participants were 3.3 years younger (*p* = 0.02) and presented a higher BMI (+1.8 kg/m^2^, *p =* 0.04) than the calibration sample participants.

### 3.1. Cut-Point Calibration

The average acceleration values (in m*g* and counts/5 s) during the middle 3 min of each activity performed in the calibration sample are presented in Table 3.

Likewise, boxplots of the absolute METs observed in each activity performed in the calibration sample are presented in Figure 1. Activities were performed in a wide MET spectrum, covering from ST (≤1.5 METs) to MVPA activities (≥3 METs).

Table 4 shows the ROC-derived thresholds, together with their sensitivity, specificity, and AUC values. AUCs varied from 0.79 to 0.89 for the hip, from 0.65 to 0.86 for the non-dominant wrist, and from 0.62 to 0.86 for the dominant wrist. In the hip, we observed higher AUCs using counts rather than using m*g* to classify ST (m*g* = 0.79 (CI_95%_: 0.75–0.82); counts = 0.89 (CI_95%_: 0.86–0.92); *p* < 0.001) and MVPA (m*g* = 0.86 (CI_95%_: 0.83–0.89); counts = 0.89 (CI_95%_: 0.87–0.92); *p* < 0.001). However, in both wrists, AUCs for the classification of ST were not significantly different (non-dominant wrist: m*g* = 0.86 (CI_95%_: 0.83–0.89), counts = 0.85 (CI_95%_: 0.82-0.89), *p* = 0.370; dominant wrist: m*g* = 0.86 (CI_95%_: 0.83–0.90), counts = 0.85 (CI_95%_: 0.82–0.88), *p* = 0.197). AUCs for the MVPA classification were higher using m*g* than using counts (non-dominant wrist: m*g* = 0.74 (CI_95%_: 0.70–0.78), counts = 0.65 (CI_95%_: 0.61–0.69), *p* < 0.001; dominant wrist: m*g* = 0.73 (CI_95%_: 0.69–0.77), counts = 0.62 (CI_95%_: 0.58–0.66), *p* < 0.001). Of note, AUCs were similar between wrists for the ST classification with either of the acceleration metrics used (all *p* > 0.652) and for MVPA using m*g* (*p* = 0.299). Regarding counts, the dominant wrist classified MVPA better than the non-dominant wrist (*p* = 0.008).

### 3.2. Cut-Point Cross-Validation

The METs observed in the cross-validation sample in each activity performed are shown in Appendix A (Figure A1, Appendix A). Table 5 shows the sensitivity and specificity reached by the ROC-derived thresholds in the cross-validation sample. We observed high sensitivity and specificity values for the ST-to-light PA thresholds in the hip and non-dominant wrist with m*g* and counts (sensitivity ≥0.86, specificity ≥0.80). For MVPA, the lower sensitivity value was observed for the hip threshold expressed in m*g* (i.e., 0.70), while the rest of the threshold showed high classification performance (sensitivity = 1.00, specificity ≥0.89). Sensitivity and specificity values for previously published cut-points based on similar body attachment sites and acceleration metrics than ours in older adults are presented in Appendix C (Table A2).

## 4. Discussion

To the best of our knowledge, this is the first study that developed accelerometer cut-points for the hip, non-dominant wrist, and dominant wrist with a consistent protocol in older adults (≥70 years old), cross-validating these cut-points in an independent sample following a different protocol of activities. Overall, the cut-points showed fair-to-good accuracy to classify ST and MVPA. This study contributes to the field by providing ST and MVPA cut-points from different wear locations in older adults (≥70 years old), based on a number of activities that characterize the older adults’ common activities. Furthermore, we provide cut-points based on ENMO, which are comparable across different accelerometers [21], as well as ActiGraph’s activity counts, which have been traditionally used in the field [8]. Additionally, this is the first study providing cut-points based on counts for wrist-worn accelerometers in older adults. The wide range of activities, the different wear locations, and the acceleration metrics used also allow for a deep understanding about what choices are better adapted to measure ST and PA intensities in older adults.

### 4.1. Wear Location: Hip, Non-Dominant Wrist, or Dominant Wrist

We placed accelerometers at the right hip, the non-dominant wrist, and the dominant wrist in this study. These body attachment sites are commonly used in population-based studies to monitor ST and PA in free-living conditions. As examples, the National Health And Nutrition Examination Survey (NHANES) collected accelerometer data from the hip and non-dominant wrist in a population-based sample of US citizens [22], and the UK biobank placed accelerometers on the dominant wrist of British adults [23]. As such, we provide relevant accelerometer cut-points to enhance the comparability across studies and to develop new studies with hip- and wrist-worn accelerometers in older adults.

Regarding ST, caution is advised as we did not consider posture (i.e., sitting, reclining, or lying) but only energy expenditure as a criterion. In this regard, a previous study on the AREA study participants developed ST-to-light PA cut-points for counts using activPAL-defined postures as a criterion [24]. If scaled up from counts/5 s to counts/min, the cut-points in this study are lower than the previously developed ones (i.e., 12 vs. 174 counts/min for the hip, 1224 vs. 1853 counts/min for the non-dominant wrist, and 2100 vs. 2303 counts/min for the dominant wrist) [24]. Cut-point differences can be explained by the different criterion used to identify ST and MVPA (i.e., postures vs. energy expenditure). ST based on energy expenditure could include some standing activities with low energy requirement, while the posture-based estimations might include sitting activities with high energy requirement. This may partially explain the lower thresholds observed in this study. Furthermore, we observed that ST classification accuracy was higher for wrists than the hip using ENMO (i.e., +7% in wrists) and vice versa for counts (i.e., +4% in hip). The 4% higher performance in the hip versus wrist counts was also observed in a previous study on the AREA study participants [24]. In addition, previous studies using ENMO have found a similar accuracy of the hip and wrist for the ST classification (i.e., 7 m*g* in this study, 12 m*g* in Duncan et al. [7], and 6 m*g* in Sanders et al. [6]) and the non-dominant wrist (i.e., 18 m*g* in this study, 18 m*g* in Duncan et al. [7], and 20 m*g* in Sanders et al. [6]). The dominant wrist threshold was higher in our study (i.e., 22 m*g*) compared with Duncan et al. (i.e., 10 m*g*) [7]. The inclusion of several sedentary activities with arm movements, (e.g., writing, dealing cards) should be considered as they can lower the performance of some ROC curves, yet they are more representative of daily life activities of older adults.

We found better accuracy to classify MVPA by hip data than wrist data in the acceleration metrics investigated (i.e., +13% to +26% for the hip compared to wrists). Two previous studies also found better accuracy in hip monitors compared to wrist monitors to classify MVPA based on ENMO [6,7]. Between-location differences were smaller in Sanders et al.’s cut-points (i.e., +6% accuracy in the hip) [6] than in Duncan et al.’s cut-points (i.e., +16% to +19% accuracy in the hip) [7]. Different protocols of activities are likely responsible of the different estimations, as well as differences in the study samples investigated (e.g., we focused on an older sample (≥70 years old) compared to previous studies). Likewise, the definition of the 1 MET value may explain how we found lower thresholds than previous studies, since we used VO_2_ = 2.8 mL/kg/min instead of the higher values (i.e., 3 to 3.5 mL/kg/min) used in some of the previous studies [6,7,16]. To note, the hip versus non-dominant wrist differences were roughly similar after removing cycling from the activities analyzed (i.e., +16% with cycling included) [7]. An overall outperformance of hip versus wrist in the classification of MVPA can be concluded from the different studies using ENMO [6,7]. This can be partially explained by the higher variability observed in wrist accelerations, which may complicate their classification into specific categories. No previous studies have provided MVPA cut-points for counts using wrist-worn accelerometers in older adults.

Overall, dominant and non-dominant wrists provided a similar classification accuracy both for ST and MVPA in this study. Similar conclusions were obtained for the ST classification with counts in a previous study on the AREA study participants [24]. Duncan et al. found a better classification accuracy of dominant wrist vs. non-dominant wrist for ST and vice versa for MVPA [7], although this depended on the activities included in the analysis (i.e., considering or not cycling) [7]. Fraysse et al. observed a similar accuracy of dominant and non-dominant wrists for the classification of ST yet a slightly higher accuracy of the non-dominant compared to the dominant wrist to classify MVPA (i.e., +5%) [25]. Of note, cut-points developed by Fraysse et al. were based on a different acceleration metric than ours; thus, the absolute thresholds cannot be directly compared.

### 4.2. Acceleration Metrics: ENMO or Counts

The inclusion of both ENMO and counts is of relevance in this study. Although ActiGraph activity counts have been traditionally used in the field, concerns have been raised about their replicability with other monitors and the lack of transparency in the processing methods to obtain them [26]. In this regard, modern accelerometers provide raw accelerations that can be consistently processed to obtain comparable outputs from different monitors [27]. In this line, open-source metrics, such as ENMO, are of great value for the field as they ease replicability and comparability across different cohorts using different devices [11,28]. We provided cut-points using both ENMO and counts in this study in order to fit different needs.

Regarding the hip cut-points, counts outperformed ENMO in the classification of ST and MVPA by 10% and 3%, respectively. Otherwise, ENMO outperformed counts in the classification of MVPA in both wrists by 9–11%. The fact that the ENMO metric was originally developed from wrist data [29], while counts were developed from hip data, can partly explain this finding. It may be that the calibration and filtering procedures of ENMO and counts are better adapted to wrist and hip motion, respectively. To our knowledge, no previous studies have compared ENMO and counts in the classification of ST and MVPA, which is a novel finding in this study.

### 4.3. Choosing Cut-Points

We provided cut-points to fit different needs in this study. A common scenario is that the data were previously collected; thus, decisions on wear location were already decided. In this case, we provided cut-points for the hip, non-dominant wrist, and dominant wrist to adapt to the different data collection protocols. Likewise, raw data require high storage capacity, hence are often removed after their initial processing. In this case, we provided cut-points based on two different acceleration metrics that are most frequently used in the field at the moment (i.e., ENMO and counts).

In a different scenario, data collection was not performed yet and advanced decisions can be made. In this regard, a smart decision would be to use cut-points with higher accuracy for the main variable of interest (e.g., ST or MVPA). Yet, other considerations apart from accuracy should be balanced. For example, acceptability of accelerometers could be higher for certain body attachment sites in a given population, resulting in higher compliance with the accelerometers [30,31]. Likewise, other accelerometer-assessed variables, such as sleep, might be of interest, and sleep algorithms are mainly based on wrist-worn accelerometers, which may point out the body attachment site to use in a hypothetical study. While not always possible, it would be of value to cross-validate the cut-points with the sample of interest (or a subsample of it). Although we cross-validated our cut-points with an independent sample, they might not be extrapolated to every older adult’s sample. Regarding our cross-validation, we roughly observed high sensitivity and specificity values for ST and MVPA in our hip and non-dominant wrist cut-points (sensitivity > 0.83 and specificity > 0.80). The ENMO cut-point for the hip requires further attention as its sensitivity was lower than that of the rest (i.e., 0.70). A similar scenario was observed in a previous study (i.e., 19 m*g*), and the authors suggested the relevance of increasing the specificity as much as possible, while keeping the sensitivity at 0.6 to ensure that light PA was not misclassified as MVPA [6]. Following this approach to define the threshold in our calibration sample, we obtained a cut-point of 23 m*g*, which is still lower than previous thresholds proposed for older adults (e.g., ~55 m*g* [7]*,* 69 m*g* [6]). The activities selected in the calibration protocol may be partially responsible of this, with some activities requiring low hip movement but light-to-moderate-intensity MET values (e.g., standing still, washing dishes, kneading dough, dressing, shopping, dealing cards). Indeed, we observed a lower MVPA cut-point for counts in the hip than previously proposed cut-points. If scaled up to 15 s epochs, we provided a cut-point of 162 counts/15 s, while Evenson et al. proposed cut-points from 296 to 620 counts/15 s in 200 older women [32]. Unfortunately, no cross-validation of the dominant wrist cut-points could be performed. Future studies should investigate the agreement between the ST and MVPA metrics obtained in free-living conditions with the different sets of cut-points presented in this paper.

### 4.4. Limitations and Strengths

This study is not without limitations. First, although we had a relatively large sample size compared to previous studies, we lost data of 30 participants (38%) because of calibration issues with the accelerometer data or missing data. Age, sex, and anthropometrics were similar between included and excluded participants. Second, cut-points are specific of the data collection protocol and participants taking part in the calibration study. To improve generalization, we cross-validated the cut-points in an independent sample performing different activities. However, this may not be enough, and more cross-validation studies should be performed in older adults. Population-standardized values could also be used to harmonize and improve representativity of accelerometer data. Likewise, our definition of 1 MET (i.e., 2.8 mL/kg/min) could not be determined specifically for each individual since our protocol did not include a sitting activity of enough duration. Instead, we calculated the oxygen consumption while lying for 10 min to support the decision previously proposed in the literature of using lower oxygen consumption values than the standard 3.5 mL/kg/min [14]. Current guidelines suggest measuring the resting metabolic rate (i.e., 1 MET) while participants are lying down, not sitting [15]. Additionally, lower values of oxygen consumption have also been used in previous calibration studies on older adults [6,16,17]. Furthermore, the accelerometers did collect raw accelerations at different frequencies in the calibration (i.e., 80 Hz) and cross-validation (i.e., 60 Hz) studies. An effect of the sampling frequency in the count generation has been previously observed [33], and this may affect the results of the cross-validation of our count-based cut-points. However, such effect was observed at relatively high intensities, which are unlikely to occur in our ≥70-year-old participants. Another limitation is that we could not cross-validate the dominant wrist cut-points. Finally, we acknowledge that the laboratory setting limited the ecological validity of the data, even if the participants performed activities that replicated their daily life activities.

The strengths of this study included the focus on ≥70-year-olds, a population with limited evidence thus far. We used 2.8 mL/min as a reference for the MET calculation, being better adapted to older populations than the standard 3.5 mL/min [14]. Cut-points were developed based on tri-axial accelerations with two different acceleration metrics and placing the accelerometers in three body attachment sites (i.e., hip, non-dominant wrist, and dominant wrist). Open-source algorithms were used for ENMO-based cut-points, which enhanced the replicability and comparability across studies. Likewise, we used a wide array of tasks with a variety of activity intensities, including activities with substantial arm movements, which increased the extrapolation to free living studies.

## 5. Conclusions

This study provided cut-points for the classification of ST and MVPA in older adults based on two different acceleration metrics (i.e., ENMO (m*g*) and ActiGraph activity counts) and three body attachment sites (i.e., hip, non-dominant wrist, and dominant wrist). The cut-points showed fair-to-good accuracy and high sensitivity and specificity values in an independent sample of participants performing different activities. Overall, ENMO-based cut-points outperformed counts in wrist-worn devices, and the opposite was observed in hip-worn devices. Hip-worn devices provided a better classification of MVPA, and the accuracy of non-dominant and dominant wrists was similar for both ST and MVPA. Further cross-validation studies and population-standardized values would benefit the field in searching for cut-points with clinical meaningfulness in older adults.

## Figures and Tables

**Figure 1 sensors-21-03326-f001:**
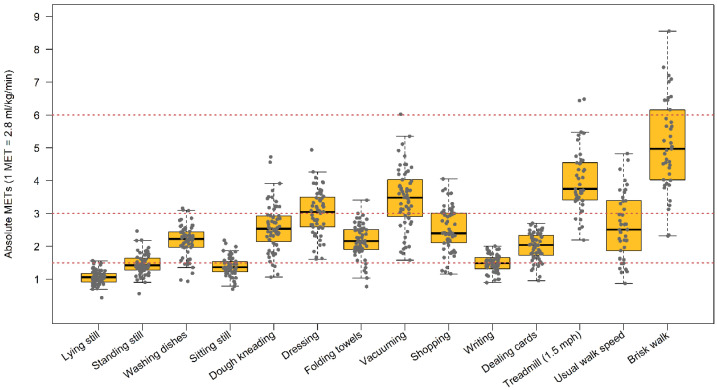
Box plot for the METs to each activity performed in the calibration sample (AREA). Red dotted lines represent the thresholds to define the different categories (i.e., 1.5, 3, and 6 METs for ST, moderate, and vigorous PA, respectively). MET: metabolic equivalent of the task.

**Table 1 sensors-21-03326-t001:** Description of the activities performed in the laboratory protocol.

Activity	Brief Description	Recorded Time (min)	Used for Acc. Data (min)	Used for METs (min)
AREA study				
Lying still	Lying supine, avoiding movement	11.1 (1.2)	3 (0)	2.0 (0.1)
Standing still	Standing upright, avoiding movement	3.2 (0.6)	2.8 (0.5)	2.0 (0.3)
Sitting still	Sitting in a chair, avoiding movement	3.1 (0.3)	2.9 (0.3)	2.0 (0.1)
Washing dishes	Standing upright, washing dishes	3.4 (0.5)	2.9 (0.2)	2.0 (0.1)
Kneading dough	Standing upright, kneading dough	3.2 (0.3)	2.9 (0.2)	2.0 (0.1)
Dressing up	Standing upright, dressing up	3.5 (0.5)	3.0 (0.1)	2.0 (0.1)
Folding towels	Standing upright, folding towels	3.1 (0.4)	2.8 (0.2)	2.0 (0.1)
Vacuuming	Standing upright, vacuuming	3.3 (0.4)	2.8 (0.2)	2.0 (0.1)
Shopping	Standing upright, shopping for groceries	3.8 (0.7)	3.0 (0.2)	2.0 (0.1)
Writing	Sitting in a chair, writing	3.8 (0.6)	3.0 (0.2)	2.0 (0.1)
Dealing cards	Sitting in a chair, dealing cards	3.2 (0.4)	2.9 (0.3)	2.0 (0.1)
Walking on a treadmill	Walking on a treadmill at 1.5 mph	5.9 (0.4)	3.0 (0.0)	2.0 (0.1)
Walking	Over-ground walking at usual speed	1.3 (0.2)	1.0 (0.2)	1.0 (0.2)
Fast walking	Over-ground brisk walking	5.6 (1.1)	3.0 (0.0)	2.0 (0.1)
MOBEMENT study		**Protocol Duration (min)**		
Reading	Sitting in a chair, reading the newspaper		5	
Watching TV	Sitting in a chair, watching TV		5	
Handcrafting	Sitting in a chair, handcrafting		5	
Standing still	Standing upright, avoiding movement		5	
Making the bed	Standing upright, making the bed		5	
Walking	Over-ground walking at usual speed		5	
Sweeping	Standing upright, sweeping and mopping		5	
Climbing stairs	Standing upright, climbing stairs		5	
Walking	Over-ground brisk walking (6 min test)		6	

Acc: accelerometer; METs: metabolic equivalents of the task.

**Table 2 sensors-21-03326-t002:** Descriptive characteristics of participants in the calibration (AREA) and cross-validation (MOBEMENT) samples.

**Calibration**	**All (*N* = 59)**	**Men (*N* = 29)**	**Women (*N* = 30)**
Age (years)	78.7 ± 5.7	78.6 ± 6.1	78.9 ± 5.3
Height (cm)	166.0 ± 8.5	172.5 ± 5.9	159.8 ± 5.2
Weight (kg)	73.3 ± 12.4	78.7 ± 11	68.2 ± 11.7
BMI (kg/m^2^)	26.5 ± 3.6	26.4 ± 3.2	26.6 ± 4.0
**Cross-Validation**	**All (*N* = 19)**	**Men (*N* = 9)**	**Women (*N* = 10)**
Age (years)	75.5 ± 4.6	75.5 ± 4.5	75.5 ± 4.9
Height (cm)	159.9 ± 8.7	166.5 ± 7.6	154.0 ± 4.4
Weight (kg)	72.6 ± 11.9	80.3 ± 11.1	65.7 ± 7.8
BMI (kg/m^2^)	28.3 ± 3.0	29.0 ± 3.3	27.6 ± 2.7

Data are presented as the mean ± standard deviation. BMI: body mass index; VO_2_: volume of oxygen consumption.

**Table 3 sensors-21-03326-t003:** Accelerations values during each laboratory activity in the calibration sample (AREA).

Activities	Acceleration Metric	Hip	Non-Dominant Wrist	Dominant Wrist
Lying still	ENMO (m*g*)	3.4 ± 5.3	6.4 ± 6.4	5.5 ± 5.5
*n* = 59	Counts/5 s	0.0 ± 0.0	2.3 ± 11.7	1.1 ± 4.8
Standing still	ENMO (m*g*)	6.7 ± 6.1	12.5 ± 13.7	13.5 ± 17.4
*n* = 59	Counts/5 s	0.0 ± 0.0	4.7 ± 11.2	5 ± 12.8
Sitting still	ENMO (m*g*)	6.0 ± 6.0	8.6 ± 8.0	8.3 ± 6.6
*n* = 59	Counts/5 s	0.0 ± 0.1	18.0 ± 34.8	16.2 ± 43.9
Washing dishes	ENMO (m*g*)	8.2 ± 5.6	71.9 ± 43.9	101.1 ± 44.7
*n* = 59	Counts/5 s	4.4 ± 6.5	591.1 ± 202.9	832.8 ± 199.5
Kneading dough	ENMO (m*g*)	10.8 ± 6.7	65.3 ± 24.2	69.1 ± 24.8
*n* = 59	Counts/5 s	26.1 ± 25.7	387.1 ± 170.8	410.1 ± 160.8
Dressing	ENMO (m*g*)	17.1 ± 6.9	120.2 ± 39.1	119.3 ± 37.4
*n* = 59	Counts/5 s	77.9 ± 41.2	867.1 ± 174.3	900 ± 170.8
Folding towels	ENMO (m*g*)	8.4 ± 5.9	85.5 ± 20.5	86.9 ± 19.3
*n* = 59	Counts/5 s	21.0 ± 20.1	728.1 ± 129.1	734.5 ± 111.8
Vacuuming	ENMO (m*g*)	25.7 ± 11.3	46.2 ± 22.4	65.2 ± 22.4
*n* = 59	Counts/5 s	174.5 ± 58.4	247.8 ± 115.3	364.6 ± 118.2
Shopping	ENMO (m*g*)	11.6 ± 5.1	49.9 ± 22.4	59.7 ± 18.0
*n* = 59	Counts/5 s	55.0 ± 40.5	503.7 ± 228.2	637.1 ± 195.3
Writing	ENMO (m*g*)	4.8 ± 4.5	12.5 ± 5.0	15.2 ± 5.4
*n* = 58	Counts/5 s	2.1 ± 4.4	117.2 ± 59.4	111.5 ± 42.7
Dealing cards	ENMO (m*g*)	7.8 ± 6.2	31.0 ± 22.0	47.1 ± 24.4
*n* = 59	Counts/5 s	20.0 ± 26.3	224.1 ± 223.7	403.5 ± 238.4
Walking on the treadmill (1.5 mph)	ENMO (m*g*)	48.4 ± 10.6	31.5 ± 25.4	33.3 ± 37.9
*n* = 41	Counts/5 s	110.5 ± 36.5	78.0 ± 81.0	92.3 ± 118.7
Usual walk speed	ENMO (m*g*)	94.3 ± 39.6	117 ± 57.8	130.3 ± 83.7
*n* = 41	Counts/5 s	245.4 ± 75.6	347.0 ± 116.4	368.0 ± 153.8
Brisk walk	ENMO (m*g*)	137.2 ± 51.8	187.7 ± 139.8	205.8 ± 161.5
*n* = 41	Counts/5 s	307.2 ± 132.1	466.9 ± 236.8	508.5 ± 269.3

Data are presented as the mean (standard deviation). ENMO: Euclidean Norm Minus One *G* with negative values rounded to zero.

**Table 4 sensors-21-03326-t004:** Derived thresholds to classify ST, light, and MVPA intensities in the calibration sample (AREA).

Site	Acceleration Metric	Threshold	Sensitivity	Specificity	AUC
**Hip**					
ST to light PA	ENMO (m*g*)	7	0.67	0.77	0.79
	Counts/5 s	1	0.84	0.85	0.89
Light to MVPA	ENMO (m*g*)	14	0.80	0.77	0.86
	Counts/5 s	54	0.84	0.82	0.89
**Non-dominant wrist**					
ST to light PA	ENMO (m*g*)	18	0.77	0.83	0.86
	Counts/5 s	102	0.77	0.82	0.85
Light to MVPA	ENMO (m*g*)	60	0.61	0.68	0.74
	Counts/5 s	182	0.77	0.50	0.65
**Dominant wrist**					
ST to light PA	ENMO (m*g*)	22	0.78	0.83	0.86
	Counts/5 s	175	0.83	0.78	0.85
Light to MVPA	ENMO (m*g*)	64	0.69	0.64	0.73
	Counts/5 s	268	0.76	0.49	0.62

AUC: area under the curve; ENMO: Euclidean Norm Minus One *G* with negative values rounded to zero; MVPA: moderate-to-vigorous physical activity; PA: physical activity; ST: sedentary time.

**Table 5 sensors-21-03326-t005:** Cross-validation of the previously derived thresholds to classify ST, light, and MVPA intensities in the cross-validation sample (MOBEMENT).

Site	Acceleration Metric	Threshold	Sensitivity	Specificity
**Hip**				
ST to light PA	ENMO (m*g*)	7	0.88	0.80
	Counts/5 s	1	0.91	0.96
Light PA to MVPA	ENMO (m*g*)	14	0.70	0.99
	Counts/5 s	54	1.00	0.96
**Non-dominant wrist**				
ST to light PA	ENMO (m*g*)	18	0.86	0.86
	Counts/5 s	102	0.91	0.92
Light PA to MVPA	ENMO (m*g*)	60	0.83	0.99
	Counts/5 s	182	1.00	0.89

ENMO: Euclidean Norm Minus One *G* with negative values rounded to zero; MVPA: moderate-to-vigorous physical activity; PA: physical activity; ST: sedentary time.

## Data Availability

The data presented in this study are available on request from the corresponding author. The data are not publicly available due to ethical reasons.

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
