# Peer review of "Calibration and Cross-Validation of Accelerometer Cut-Points to Classify Sedentary Time and Physical Activity from Hip and Non-Dominant and Dominant Wrists in Older Adults"

_sensors, 2021, doi:10.3390/s21103326_

Round 1

Reviewer 1 Report

In this study, the authors tried to develop the cut-off points of accelerometer wearing on the waist and wrist in older adults. The manuscript was well written, and the reviewer would like to acknowledge points of interest in the study; however, there are significant concerns that should be improved as follows:

Major:

  • The reviewer quite agrees with using 3.5mL/kg/min as 1MET. However, generally (at least to my knowledge), 1 MET is defined as resting energy expenditure during sitting still, but not lying down. Thus, there would not be any telling argument, the reviewer basically recommends re-calculation of all METs values by using energy expenditure during sitting. Probably, according to the results of the mean METs values during sitting still, mean 1 MET value should be closer to 3.5 ml/kg/min. Also, naturally, because of this re-calculation, please revise the Discussion section.
  • In this study, the “closest to (0,1)” method was likely used for evaluating the cut-off points. Why did the authors the method, but not the Youden index? Using the Youden index, the cut-off points and the results of the study changed? Please explain (discuss, if needed) this point.
  • Using the other cut-off points for the Actigraph, provided by the previous studies as the authors indicated, for this sample, how much does the AUC change? If possible, please consider showing the results as a piece of the manuscript.

Minor comments:

  • Line 149,172, 220, there is no specific unit for METs. Thus, please remove the unit (ml/kg/min) from the manuscript.
  • For the assessment of raw accelerations, the spec (e.g. resolution, etc) of the sensor (device) should be important. The ActiGraph is actually familiar to researchers in the physical activity field; however, please describe the sensor specification in detail for the broad readers.

Author Response

------------------------------------------------------------

REVIEWER 1

------------------------------------------------------------

Comment #1: In this study, the authors tried to develop the cut-off points of accelerometer wearing on the waist and wrist in older adults. The manuscript was well written, and the reviewer would like to acknowledge points of interest in the study; however, there are significant concerns that should be improved as follows:

Response #1: We appreciate your kind remarks.  Please find our answers to your suggestions below.

Comment #2: The reviewer quite agrees with using 3.5mL/kg/min as 1MET. However, generally (at least to my knowledge), 1 MET is defined as resting energy expenditure during sitting still, but not lying down. Thus, there would not be any telling argument, the reviewer basically recommends re-calculation of all METs values by using energy expenditure during sitting. Probably, according to the results of the mean METs values during sitting still, mean 1 MET value should be closer to 3.5 ml/kg/min. Also, naturally, because of this re-calculation, please revise the Discussion section.

Response #2: We agree that 1 MET is defined as the metabolic rate while sitting in a resting status. Among our activities, lying still is the most similar one with enough duration as to roughly determine the value of 1 MET. Furthermore, a previous study did not observe differences between lying and sitting (i.e., <2% difference, not statistically significant) [1]. However, as our methodology was not ideal (i.e., only 10-min collected while lying – not sitting), we decided not to use these individualized MET values and rather use this information as an argument to support our decision of using a lower oxygen consumption value than the standard 3.5 ml/kg/min. This decision is further supported by previous evidence as we explain in the manuscript [2]; and previous studies developing cut-points in older adults have used lower values for the definition of 1 MET than the standard 3.5 ml/kg/min [3–5]. In this regard, the current indirect calorimetry guidelines suggest that the ‘real’ resting metabolic rate (i.e., 1 MET) should be obtained with participants lying down, not sitting [6]. As such, we would rather keep our original plan of defining 1 MET as 2.8 ml/kg/min. We have specified this as a limitation in the manuscript (lines 398-404): “Likewise, our definition of 1 MET (i.e., 2.8 ml/kg/min) could not be determined specifically for each individual since our protocol did not include a sitting activity of enough duration. Instead, we calculated the oxygen consumption while lying for 10 min to support the decision previously proposed in literature of using lower oxygen consumption values than the standard 3.5 ml/kg/min [14]. Current guidelines suggest measuring the resting metabolic rate (i.e., 1 MET) while participants are lying down, not sit-ting [15]. Additionally, lower values of oxygen consumption have also been used in previous calibration studies in older adults [6,26,27].”

Comment #3: In this study, the “closest to (0,1)” method was likely used for evaluating the cut-off points. Why did the authors the method, but not the Youden index? Using the Youden index, the cut-off points and the results of the study changed? Please explain (discuss, if needed) this point.

Response #3: As sensitivity analyses, we also calculated cut-points with the Youden index. Most of the cut-points were similar, yet some of the cut-points for MVPA were substantially lower. The same was found in a previous study in which cut-points derived with these two different approaches were similar for ST, yet they differed for MVPA [4]. Our values using the “closest to (0,1)” were more realistic, closer to previous research, and resulted in higher sensitivity and specificity values in the cross-validation analyses. Thus, we decided to present these cut-points. Find here the cut-points using the Youden index in Table R1:

Table R1. Derived thresholds to classify ST, light, and MVPA intensities in the calibration sample using the Youden index (AREA).

Site

Acceleration Metric

Threshold

Sensitivity

Specificity

AUC

Hip

ST to light PA

ENMO (mg)

7

0.67

0.77

0.79

Counts/5s

1

0.84

0.85

0.89

Light to MVPA

ENMO (mg)

12

0.87

0.70

0.86

Counts/5s

35

0.90

0.77

0.89

Non-dominant wrist

ST to light PA

ENMO (mg)

17

0.77

0.83

0.86

Counts/5s

72

0.73

0.86

0.85

Light to MVPA

ENMO (mg)

18

0.93

0.41

0.74

Counts/5s

129

0.85

0.45

0.65

Dominant wrist

ST to light PA

ENMO (mg)

22

0.78

0.83

0.86

Counts/5s

175

0.83

0.78

0.85

Light to MVPA

ENMO (mg)

56

0.76

0.58

0.73

Counts/5s

196

0.83

0.46

0.62

AUC: area under the curve, ENMO: Euclidean Norm Minus One G with negative values rounded to zero, MVPA: moderate-to-vigorous physical activity, PA: physical activity, ST: sedentary time.

Comment #4: Using the other cut-off points for the Actigraph, provided by the previous studies as the authors indicated, for this sample, how much does the AUC change? If possible, please consider showing the results as a piece of the manuscript.

Response #4: AUC values do not change as a function of the cut-points. AUC is firstly calculated and then cut-points are decided over the curve of sensitivity vs specificity values arising for each cut-point tested in the ROC curve. Having said that, we did calculate the sensitivity and specificity values obtained from previous studies using accelerometer metrics similar to ours. We used the data from our cross-validation sample for this purpose. We have included this information in the appendix C (Table C1).

Table C1. Cross-validation of the previously-derive thresholds to classify ST, light, and MVPA intensities in the cross-validation sample (MOBEMENT).

Reference/Cut-point

Site

Acceleration metric

Threshold

Sensitivity

Specificity

Evenson et al.1 [4]

ST to light PA

Hip

Counts/5s

1

0.88

0.80

Light PA to MVPA

Hip

Counts/5s

27 

1.00

0.96

Evenson et al.2 [4]

ST to light PA

Hip

Counts/5s

6

0.97

0.91

Light PA to MVPA

Hip

Counts/5s

64 

1.00

0.96

Duncan et al. 1 [7]

ST to light PA

Hip

ENMO (mg)

12

0.98

0.64

Non-dominant wrist

ENMO (mg)

18

0.86

0.86

Light PA to MVPA

Hip

ENMO (mg)

55

0.57

1.00

Non-dominant wrist

ENMO (mg)

122

0.48

0.99

Sanders et al.1 [3]

ST to light PA

Hip

ENMO (mg)

6

0.81

0.82

Non-dominant wrist

ENMO (mg)

20

0.86

0.85

Light PA to MVPA

Hip

ENMO (mg)

19

0.63

1.00

Non-dominant wrist

ENMO (mg)

32

0.94

0.94

Sanders et al.3 [3]

ST to light PA

Hip

ENMO (mg)

15

1.00

0.58

Non-dominant wrist

ENMO (mg)

57

0.97

0.75

Light PA to MVPA

Hip

ENMO (mg)

69

0.51

1.00

Non-dominant wrist

ENMO (mg)

104

0.61

0.99

ENMO: Euclidean Norm Minus One G with negative values rounded to zero, MVPA: moderate-to-vigorous physical activity, PA: physical activity, ST: sedentary time.

1 Cut-points derived maximizing the sum of sensitivity plus specificity (Youden index).

2 Cut-points derived balancing the number of false positives and false negatives (top-left approach).

3 Cut-points derived increasing sensitivity for ST and specificity for MVPA.

Comment #5: Line 149,172, 220, there is no specific unit for METs. Thus, please remove the unit (ml/kg/min) from the manuscript.

Response #5: We have made this correction throughout the manuscript. Thank you.

Comment #6: For the assessment of raw accelerations, the spec (e.g. resolution, etc) of the sensor (device) should be important. The ActiGraph is actually familiar to researchers in the physical activity field; however, please describe the sensor specification in detail for the broad readers.

Response #6: Device specifications are now included in lines 123-125: “The GT3X+ is a watch-sized light-weight device able to collect accelerations at 30-100 Hz within a dynamic range of ±8 G’s in the three axes: vertical, antero-posterior, and medio-lateral axes”.

References

  1. Miles-Chan, J.L.; Sarafian, D.; Montani, J.P.; Schutz, Y.; Dulloo, A.G. Sitting comfortably versus lying down: Is there really a difference in energy expenditure? Clin. Nutr. 2014, 33, 175–178.
  2. Kwan, M.; Woo, J.; Kwok, T. The standard oxygen consumption value equivalent to one metabolic equivalent (3.5 ml/min/kg) is not appropriate for elderly people. Int. J. Food Sci. Nutr. 2004, 55, 179–182.
  3. Sanders, G.J.; Boddy, L.M.; Sparks, S.A.; Curry, W.B.; Roe, B.; Kaehne, A.; Fairclough, S.J. Evaluation of wrist and hip sedentary behaviour and moderate-to-vigorous physical activity raw acceleration cutpoints in older adults. J. Sports Sci. 2019, 37, 1270–1279.
  4. Evenson, K.R.; Wen, F.; Herring, A.H.; Di, C.; LaMonte, M.J.; Tinker, L.F.; Lee, I.M.; Rillamas-Sun, E.; LaCroix, A.Z.; Buchner, D.M. Calibrating physical activity intensity for hip-worn accelerometry in women age 60 to 91years: The Women’s Health Initiative OPACH Calibration Study. Prev. Med. Reports 2015, 2, 750–756.
  5. Fraysse, F.; Post, D.; Eston, R.; Kasai, D.; Rowlands, A. V.; Parfitt, G. Physical Activity Intensity Cut-Points for Wrist-Worn GENEActiv in Older Adults. Front. Sport. Act. Living 2021, 2, 1–9.
  6. Fullmer, S.; Benson-Davies, S.; Earthman, C.P.; Frankenfield, D.C.; Gradwell, E.; Lee, P.S.P.; Piemonte, T.; Trabulsi, J. Evidence Analysis Library Review of Best Practices for Performing Indirect Calorimetry in Healthy and Non-Critically Ill Individuals. J. Acad. Nutr. Diet. 2015, 115, 1417-1446.e2.
  7. Duncan, M.J.; Rowlands, A.; Lawson, C.; Leddington Wright, S.; Hill, M.; Morris, M.; Eyre, E.; Tallis, J. Using accelerometry to classify physical activity intensity in older adults: What is the optimal wear-site? Eur. J. Sport Sci. 2019, 0, 1–27.

Reviewer 2 Report

Dear authors

This was a very good example of a classical calibration and validation study, with a very well-written manuscript. It was a pleasure to read it. I have only minor comments.

First, I observed that the sampling frequency of 80Hz was used in the calibration study and 60Hz in the cross-validation study. My concern when using the ActiLife algorithm to produce counts, is that the calibration study could suffer from the sampling frequency error with 80Hz sampling frequency not found with the 60Hz sampling frequency (Brønd & Arvidsson 2016). Although, the activity intensity would not be that high in this age-group, the effect may be present. In the calibration study the sensitivity and specificity was lower than for the cross-validation study, which may also be explained by a random error variation with 80Hz not seen with 60Hz. Please check whether the inter-individual variation in counts for each separate activity in the calibration study differed considerably compared to the inter-individual variation in the cross-validation study.

Table 1: Please add the time unit for recorded time in the AREA study.

Page 5: First paragraph: Please provide a reference stating that cut-points for SED and MVPA corresponds to 1.5ml/kg/min and 3.0ml/kg/min also in adults 70+. Should 1.5 and 3.0 be used as cut-points for LPA and MPA, respectively, in this age-group like in younger age-groups?

Page 11, row 377: sentence “Indeed, a lower MVPA…” is incomplete, please revise.

Author Response

------------------------------------------------------------

REVIEWER 2

------------------------------------------------------------

Comment #1: This was a very good example of a classical calibration and validation study, with a very well-written manuscript. It was a pleasure to read it. I have only minor comments.

Response #1: Thank you for your appreciations.

Comment #2: First, I observed that the sampling frequency of 80Hz was used in the calibration study and 60Hz in the cross-validation study. My concern when using the ActiLife algorithm to produce counts, is that the calibration study could suffer from the sampling frequency error with 80Hz sampling frequency not found with the 60Hz sampling frequency (Brønd & Arvidsson 2016). Although, the activity intensity would not be that high in this age-group, the effect may be present. In the calibration study the sensitivity and specificity was lower than for the cross-validation study, which may also be explained by a random error variation with 80Hz not seen with 60Hz. Please check whether the inter-individual variation in counts for each separate activity in the calibration study differed considerably compared to the inter-individual variation in the cross-validation study.

Response #2: The laboratory protocols in the calibration and cross-validation studies were different and so the activities were. Thus, we cannot make strong conclusions about whether the interindividual differences across protocols are due to the different activities performed or the sampling frequencies. However, we do agree with the reviewer that the different sampling frequencies may affect the output and we have included this as a limitation (lines 404-409): “Furthermore, the accelerometers did collect raw accelerations at different frequencies in the calibration (i.e., 80 Hz) and cross-validation (i.e., 60 Hz) studies. An effect of the sampling frequency in the counts generation has been previously observed [1], and this may affect the results of the cross-validation of our count-based cut-points. However, such effect was observed at relatively high intensities, which are unlikely to occur in our >70-year-old participants”.

Comment #3: Table 1: Please add the time unit for recorded time in the AREA study.

Response #3: Done, thank you.

Comment #4: Page 5: First paragraph: Please provide a reference stating that cut-points for SED and MVPA corresponds to 1.5ml/kg/min and 3.0ml/kg/min also in adults 70+. Should 1.5 and 3.0 be used as cut-points for LPA and MPA, respectively, in this age-group like in younger age-groups?

Response #4: As we adapt the value of 1 MET for this population, i.e., 2.8 ml/kg/min instead of 3.5 ml/kg/min (see response to comment #2 of reviewer #1 for more details), we can use the standard 1.5 and 3 METs to classify ST and MVPA. Previous studies in older adults have used this same approach [2–4]. This is now specified in the manuscript (lines 174-177): “METs values were used to classify each activity as ST (≤1.5 METs) or MVPA (≥3 METs) as it is standard in older adults when the value of 1 MET has been adapted to this population [6,16,17].”

Comment #5: Page 11, row 377: sentence “Indeed, a lower MVPA…” is incomplete, please revise.

Response #5: This is now revised: “Indeed, we observed a lower MVPA cut-point for counts in hip than previously proposed cut-points”.

References:

  1. Brønd, J.C.; Arvidsson, D.; Brond, J.C.; Arvidsson, D. Sampling frequency affects the processing of Actigraph raw acceleration data to activity counts. J. Appl. Physiol. 2015, 120, 362–369.
  2. Evenson, K.R.; Wen, F.; Herring, A.H.; Di, C.; LaMonte, M.J.; Tinker, L.F.; Lee, I.M.; Rillamas-Sun, E.; LaCroix, A.Z.; Buchner, D.M. Calibrating physical activity intensity for hip-worn accelerometry in women age 60 to 91years: The Women’s Health Initiative OPACH Calibration Study. Prev. Med. Reports 2015, 2, 750–756.
  3. Fraysse, F.; Post, D.; Eston, R.; Kasai, D.; Rowlands, A. V.; Parfitt, G. Physical Activity Intensity Cut-Points for Wrist-Worn GENEActiv in Older Adults. Front. Sport. Act. Living 2021, 2, 1–9.
  4. Sanders, G.J.; Boddy, L.M.; Sparks, S.A.; Curry, W.B.; Roe, B.; Kaehne, A.; Fairclough, S.J. Evaluation of wrist and hip sedentary behaviour and moderate-to-vigorous physical activity raw acceleration cutpoints in older adults. J. Sports Sci. 2019, 37, 1270–1279.

Round 2

Reviewer 1 Report

Thank you for providing answers and modifications.
Just one thing for the next studies, I believe that we need to have an ongoing discussion on whether 1MET should be defined as genuine RMR. Because most of the activities act in an upright position. So, the association between energy expenditure and acceleration should be clear linear when using energy expenditure during sitting as 1MET than during lying.